# A framework to predict the price of energy for the end-users with applications to monetary and energy policies

Stefanos G. Baratsas [1,2], Alexander M. Niziolek[1,2], Onur Onel[1,2], Logan R. Matthews[1,2], Christodoulos A. Floudas[1,2], Detlef R. Hallermann[3], Sorin M. Sorescu[3] & Efstratios N. Pistikopoulos [1,2✉]

Energy affects every single individual and entity in the world. Therefore, it is crucial to precisely quantify the "price of energy" and study how it evolves through time, through major political and social events, and through changes in energy and monetary policies. Here, we develop a predictive framework, an index to calculate the average price of energy in the United States. The complex energy landscape is thoroughly analysed to accurately determine the two key factors of this framework: the total demand of the energy products directed to the end-use sectors, and the corresponding price of each product. A rolling horizon predictive methodology is introduced to estimate future energy demands, with excellent predictive capability, shown over a period of 174 months. The effectiveness of the framework is demonstrated by addressing two policy questions of significant public interest.

[1] Artie McFerrin Department of Chemical Engineering, Texas A&M University, College Station, TX 77843, USA. [2] Texas A&M Energy Institute, Texas A&M University, College Station, TX 77843, USA. [3] Department of Finance, Mays Business School, Texas A&M University, College Station, TX 77843, USA. ✉email: stratos@tamu.edu

Energy markets are sensitive and volatile to technological breakthroughs and innovations, changes in monetary and fiscal policies, major global events, and consumer trend changes[1–3]. Various governmental agencies, and political and commercial organizations think tanks, as well as researchers and academics worldwide, consider various energy policies and their effects when dealing with the increasing concerns in energy independence, energy scarcity, energy sustainability, and pollution caused by the utilization of energy[4–9]. Furthermore, with strategic political and commercial decisions, and policies being assessed in economic terms, it is of utmost importance to accurately determine the price of energy so as to evaluate their effectiveness. Undoubtedly, energy affects every person and entity. Therefore, it is essential to accurately quantify "the price of energy" and grasp how it is affected by major breakthroughs, political events, as well as energy and monetary policies.

Given the absence of such a pre-existing tool, we introduce a predictive framework, the Energy Price Index (EPIC), which can be used as a benchmark to calculate the average price of energy to the end-use consumers in the United States. The complex energy landscape of the United States is carefully analyzed to determine the products that are directed to the end-use sectors of the US economy. The total energy demand of these products, together with their prices, serve as the backbone of EPIC. We also introduce a rolling horizon model that uses data from the past so as to estimate the weights of the energy demand in the future based on which various policy rules and questions can be assessed, designed, and optimized through the use of EPIC. The predictability of the proposed methodology is rigorously tested over a long period of 174 months, demonstrating remarkable accuracy.

The novelty of the proposed index (EPIC) lies on its unique features. First, it represents the average price of energy in the United States over the entire energy landscape covering all the different energy sources and feedstocks (non-renewables and renewables), as well as the end-use sectors. As such, it is not just a representative price of a sub-section of the energy landscape such as the price of electricity in the residential sector or the price of oil products in the industrial sector, which is common in the existing literature[10–13]. Second, the proposed formulation collectively captures the two key attributes of energy, the supply and demand mechanisms along with the prices of the energy feedstocks and products across the entire energy landscape. This is another unique feature, as the methodologies in the literature generally focus on specific energy sectors[14–16]. Third, the excellent forecasting ability of the proposed mathematical framework allows the estimation of the current value of EPIC and thus the current price of energy, overcoming the issue of the non-availability of actual data, while it can also be used to forecast future values of the energy demand accurately. In contrast, the forecasting frameworks in the literature focus on specific energy sectors[17] such as electricity[18–20], natural gas[21], crude oil[22], and petroleum products[23], with generally much shorter forecasting horizon[17–23]. Finally, it is a quantitative approach to evaluate, design, and optimize different policy questions of significant public interest such as a policy case study for the renewable energy and a policy case study for the crude oil tax.

The EPIC framework demonstrates novel features not only for the existing academic literature but also for the financial tools in the energy sector. The various energy indices are primarily capitalization weighted indices, capped market capitalization indices, price-weighted indices, and world production-weighted indices[24] with some of the most representative examples being S&P 500 Energy Index[25], MSCI US IMI Energy 25/50 Index[26], and S&P GSCI Energy[27]. S&P 500 Energy Index tracks the market capitalization of energy companies and their stock price, whereas MSCI US IMI Energy 25/50 Index captures the large,

mid, and small cap segments of the US equity universe, but neither of them reflects to the actual energy products. The S&P GSCI Energy includes only the futures contracts on physical commodities with the weights being calculated based on the world production of these commodities; however, it focuses exclusively on oil and gas products, without capturing other energy feedstocks such as electricity, or renewables. More details about these financial indices, their constituents, and their weights are shown in the Supplementary Table 1. On the contrary, EPIC captures the prices of all energy feedstocks, while the weights are calculated from the actual demands of these energy feedstocks (Supplementary Tables 2 and 3).

Two key applications of the proposed index in addressing contemporary policy questions are presented here. In particular, the effects of a crude oil tax on EPIC are investigated parametrically for a range of taxes from $2.5 per barrel up to $25 per barrel so as to estimate the expected change in the price of energy under different scenarios both for the past, i.e., what would have happened, as well as for the future, i.e., what will happen. Also, the generated revenue from the implementation of each scenario is calculated. Moreover, the effects of renewable energy production targets and subsidies on energy consumers are examined parametrically over a wide range of different weights of the energy feedstocks, as well as for tax credits ranging from 0 to 9 $/MMBtu. Similar to the previous policy case study, different scenarios are presented retrospectively and prospectively, along with the budget required for the implementation of each scenario, taking advantage of the powerful predictive ability and flexibility of the proposed methodology.

## Results and discussion
The US energy landscape is a complex and extensive network of energy feedstocks and products across multiple sectors. This complexity is due to the fact that the various energy feedstocks can be utilized in many different ways. More specifically, they can be directed straight to the end-use sectors, or converted and refined to be directed to the end-use sectors and/or to the intermediate energy-consuming sector, or directed to the intermediate energy-consuming sector. Over the years, we have extensively worked in the development of process superstructures, energy supply chain analyses, and strategic planning frameworks[28–62], which utilize single and hybrid energy feedstocks (biomass, coal, natural gas, and municipal solid waste) to produce liquid fuels and chemicals, as well as in reviewing and assessing[63–67] the current state of energy technologies, so that we obtain the required familiarity and comprehension of such a complex energy landscape.

The requirement for energy as an input to provide products and/or services is defined as energy demand[68]. As some of the energy feedstocks can be directed to the end-use sectors, the term products in this context refers to the components sent to the end-use sectors, including the primary energy sources, e.g., natural gas, coal etc. The components that are directed to the end-use sectors should be delineated, ensuring that all energy demand is accounted for and avoiding any double counting of any energy demand. It is of utmost importance to maintain a holistic and concrete approach in defining and counting the various energy products so as to be precise and consistent throughout this context. This is essential, as the total demand of the energy products directed to the end-use sectors along with their respective prices constitute the cornerstone of EPIC.

The US Energy Information Administration (EIA)[68,69] defines the energy-consuming end-use sectors as the residential, commercial, industrial, and transportation sectors of the economy, because they purchase or produce energy for their own

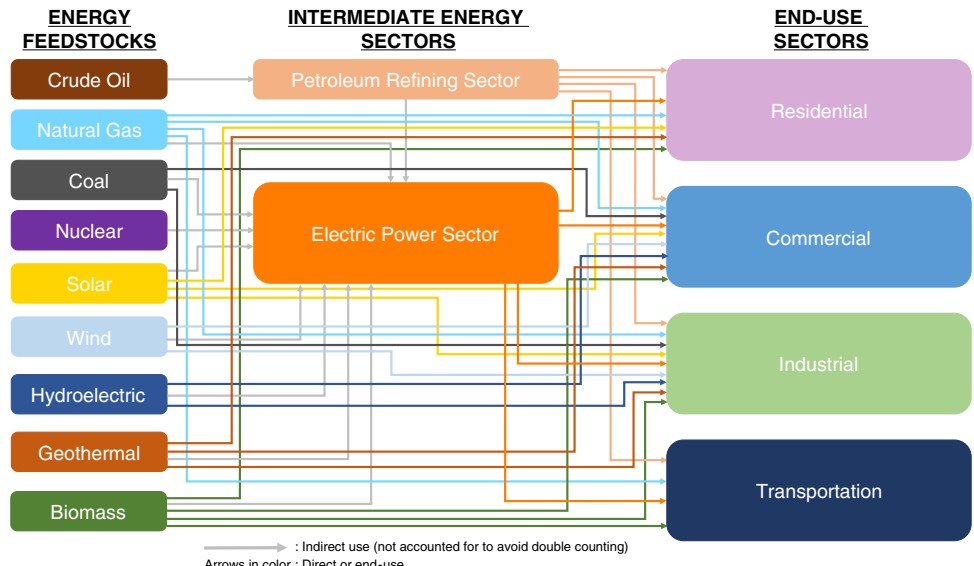

**Fig. 1 US energy landscape by source and end-use sector.** Each energy feedstock (source) has a unique color for easy visualization of the different pathways. Arrows connect energy feedstocks with the sectors that are consumed in. A gray arrow represents indirect use of an energy feedstock in an intermediate energy sector. A colored arrow (other than gray), represents direct use, matches with the color of its corresponding energy feedstock, and directs to the end-use sector that this energy feedstock is consumed in. Source: EIA[69].

consumption and not for resale. The electric power sector is defined as an intermediate energy-consuming sector, which provides electricity to the four major energy sectors, i.e., residential, commercial, industrial, and transportation[68,69]. The definitions of each of the end-use sectors can be found in Supplementary Note 1.

Figure 1 illustrates in detail the complete US energy landscape for the different energy feedstocks by source and end-use sector. Moreover, the energy landscape of each energy feedstock is shown in Supplementary Fig. 1a–e. As mentioned above, to avoid double counting, the feedstocks directed into the electric power sector do not directly enter into the EPIC calculation, because electricity is sold from the electric power sector as a product to the four end-use sectors. Therefore, the arrows going into the electric power sector are not counted, whereas the arrows leaving the electric power sector are counted.

**EPIC framework**. The two key factors comprising EPIC are the total demand of the energy products that are directed to the end-use sectors in the United States along with their respective prices.

Energy products consumed by the US economy originate from crude oil, natural gas, coal, nuclear, solar, wind, hydroelectric, geothermal, and several types of biomass. The exact determination of these products, their consumption, and their monthly prices is crucial. The full list of these energy products is presented in the Supplementary Table 2. The monthly consumption (in energy units) along with the monthly price (in $ per energy unit) for each of these energy products is obtained from the EIA and from other sources. Our data sources are presented in Supplementary Table 4. Please note that the proposed framework is generic and can be applied to (a) the United States on a national level, (b) to United States on a state-by-state basis, (c) regional level of multi-states, and (d) other countries, provided that a thorough analysis of the specific energy landscape has been conducted, the particular energy feedstocks and products have been identified, and data for their prices and demands are available.

EPIC represents the average price of energy in a given month and as such is defined as the summation of the price (in

$/MMBtu) of each product multiplied by the weight fraction of the demand of each product. The unit of EPIC is $/MMBtu. The mathematical formulation is presented in detail in the "Methods" section, whereas the necessary steps for calculating EPIC are shown in the next sections.

Figure 2 illuminates the values of EPIC from January 2003 to June 2020 in $/MMBtu.

**Rolling horizon methodology**. The weights of the demand of the various energy products become available with a lag of two to three months. In addition, the future weights of these energy products are also necessary for the evaluation, design, and optimization of policy decisions. Therefore, a predictive framework is required to estimate the present and future weights of the underlying energy products that enter into the EPIC calculation. Consequently, we propose a rolling horizon methodology as a forecasting framework to predict the values of the required data for the time period of interest, the actual values of which will not be known until a few months later. The proposed methodology is using information from the previous three time periods, so as to predict the values for the time period of interest.

Figure 3 illustrates the general concept of the rolling horizon methodology, along with its application over two stages in the future. Data from the three previous periods (T − 3, T − 2, and T − 1) are used to predict the data of interest in the current stage T. Subsequently, data from the periods T − 2, T − 1, and T are used for predicting the data of interest in stage T + 1, and so on. Additional details, along with the mathematical optimization model employed, are presented in the "Methods" section.

**Prediction of energy products' demand weights**. The rolling horizon methodology presented in the previous section is used to predict the weights of the demand of energy products for the time of interest. As the data of the energy demand lags 2–3 months, as of October 2020 we have available actual data until June 2020. To estimate the weights of energy products for July 2020, we use the data of July 2017, July 2018, and July 2019. Similarly, for August 2020, we use the data of August 2017, August 2018, and August 2019. This methodology can be extended to predict future data:

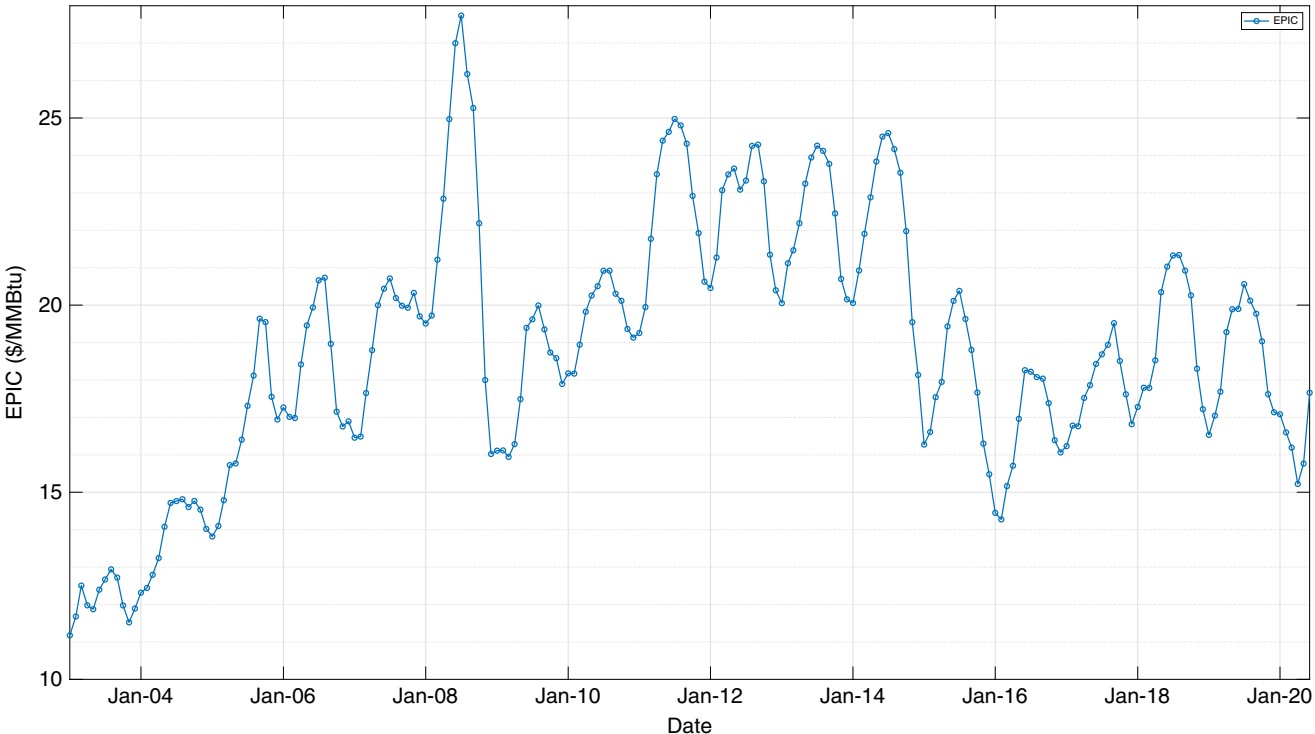

**Fig. 2 Energy Price Index - EPIC in $/MMBtu from January 2003 to June 2020 with monthly indexing.** EPIC represents the average price of energy in a given month across the United States, considering the demands and prices of all the energy feedstocks. Source data are provided as a Source Data file.

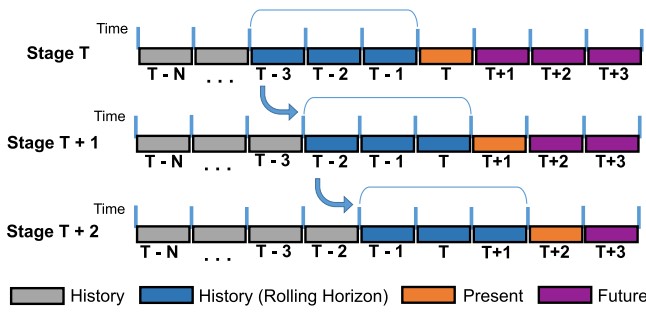

**Fig. 3 Rolling horizon methodology with application over two stages in the future.** At stage T, data from the three previous periods (T − 3, T − 2, T − 1) (blue color) are used to predict the data of interest at the current stage T (orange color). At stage T + 1, data from the periods T − 2, T − 1, T (blue color) are used for predicting the data of interest at the new present stage T + 1 (orange color). At stage T + 2, data from the periods T − 1, T, T + 1 (blue color) are used for predicting the data of interest at the new present stage T + 2 (orange color).

- 2nd year predictions require the deterministic data from the last 2 years along with the predicted data of the first year;
- 3rd year predictions require the deterministic data from the last 1 year along with the predicted data of the first and second years;
- 4th year predictions require the predicted data of the first, second, and third years.

Supplementary Fig. 2 illustrates the rolling horizon framework for the first-, second-, third-, and fourth-year prediction of the weights using September 2020 as an example.

**Weight prediction results.** The validity of our proposed methodology is tested over a period of 174 months from January 2006 to June 2020, by comparing the predicted value of the monthly

**Table 1 Weight prediction results up to 4 years from January 2006 to June 2020.**

| Year prediction | Months to compare | Average sum of squares error | Minimum sum of squares error | Maximum sum of squares error |
|---|---|---|---|---|
| 1st year | 174 | 0.000354 | 0.000050 | 0.005387 |
| 2nd year | 162 | 0.000436 | 0.000075 | 0.005242 |
| 3rd year | 150 | 0.000499 | 0.000123 | 0.006685 |
| 4th year | 138 | 0.000559 | 0.000172 | 0.006036 |

weight of each product's demand with its actual, known value. For this comparison, the sum of the squared prediction error for each month is computed over the testing period (see "Methods") and the results in the form of an average sum of squared error, minimum sum of squared error, and maximum sum of squared error are summarized in Table 1. It should be noted that the number of months to be compared decreases as the year of prediction increases. For example, the predictions of the second year require the predicted weights of the first year, so there are less actual monthly values to compare.

As shown in Table 1, the predictive ability of our proposed methodology is quite remarkable since the reported error values are extremely low. This is true even if we consider the square root of the average sum of the squared errors which is 1.8808%, 2.0874%, 2.2329%, and 2.3641% for the first, second, third, and fourth year, respectively. The very low predictive error (2.3641%) in the case of the fourth year, where only unknown (predicted) values have been used, is of significant importance. As expected, the average sum of the squared error increases, as the year of prediction increases due to the decreasing number of months with known values. The accuracy of the proposed methodology is also verified from the fact that even the maximum sum of squared error over the tested period is rather low, i.e., 0.006685 or

8.1764% when we consider the square root of the sum of the squared errors.

This excellent predictive ability along with the unique inherent characteristics of EPIC that captures both the demands and the prices of the products over the entire energy landscape in the United States justifies our opinion that EPIC is the ideal tool for designing, assessing, and optimizing various policy decisions of public interest. Two prime, representative policy case studies are presented in the next sections.

**Policy case study 1: crude oil**. The proposed framework can be used for a wide range of policy questions and analyses, and a potential tax in crude oil is investigated here as an alternative policy for mitigating climate change and concurrently generating substantial revenue that is required for climate finance. Such a policy of $10.25 per barrel tax on crude oil was also proposed by US President Barack Obama back in 2016, to support new transportation systems designed to reduce carbon emissions and congestion[70].

Here we examine parametrically the effects of a crude oil tax ranging from $2.5 per barrel up to $25 per barrel in EPIC, and calculate the amount of revenue that could be generated by such a policy from January 2003 until June 2020. Taking advantage of the excellent predictive ability of the proposed EPIC framework, we estimate the changes in EPIC in the next 4 years along with the future revenue that will be generated by such a policy. We assume that crude oil has a heating content of 5.721 MMBtu per barrel[69] and a petroleum refinery efficiency of 90%. Moreover, the amount of petroleum and petroleum products being sent for electricity generation is assumed to be negligible (≈0.57% over the last year). Crude oil demand is considered as inelastic in the short run, with long-run values of elasticity being generally higher in absolute values but still well below 1[71–74].

The average monthly difference (in $/MMBtu) and the average monthly percentage increase of EPIC in comparison to its reference values from January 2003 to June 2020 are presented in Table 2 for the different values of crude oil tax. As crude oil is inelastic, the investigated crude oil tax has not affected the historical values of crude oil consumption. As can be seen, a $10.25 per barrel of crude oil tax increases EPIC by $1.019 per MMBtu or 5.60%, whereas a $25 per barrel of crude oil tax rises EPIC by $2.484 per MMBtu or 13.66%. Table 2 also summarizes the amount of revenue generated from the investigated crude oil tax scenario over the same period (January 2003–June 2020). The

average annual revenue from a $10.25 per barrel of crude oil tax is estimated to be $70.962 billion or $17.308 billion for every $2.5 per barrel rise of crude oil tax. Supplementary Fig. 4 illustrates the recalculated EPIC for the above-mentioned values of crude oil tax along with the reference value of EPIC without tax for easy comparison, over the same period (January 2003 to June 2020).

We can also calculate the average increase in energy-related expenses per household as a result of the proposed increase in crude oil tax using data from the relevant survey published by EIA[75] in conjunction with the EPIC findings from our previous analysis. According to the 2015 survey data, the annual energy consumption per household is 77.1 MMBtu, while in 2015 the average value of EPIC is $18.01/MMBtu. Using this information, we estimate the average annual energy-related expenses per household for 2015 to be $1389.06. Taking into consideration the effects on EPIC from the increase in the crude oil taxation, a $2.5 per barrel increase in crude oil tax would have led to an average rise of $0.2432/MMBtu or 1.35% in EPIC for 2015. Therefore, the projected average annual energy related expenses per household would have increased by $18.76, up to a total of $1407.82. Similarly, an increase of $10.25 per barrel in crude oil tax would have burdened the average energy related expenses per household by $76.90 or 5.54%, up to a total of $1465.95.

The effects on EPIC from the investigated policy during the next 4 years are demonstrated in Fig. 4, using the predictive weights of demand of the energy products for this period. Similarly, with the results for the past period, the increase of EPIC in the future is investigated parametrically for different values of the crude oil tax.

According to Fig. 4, a $10.25 per barrel of crude oil tax raises EPIC over the next four years by $0.977/MMBtu on average, whereas a $25 per barrel of crude oil tax surges EPIC by $2.384/MMBtu on average in the same period. Using the weights of the demand of the petroleum energy products that have been estimated from the EPIC framework along with the projections for the total annual energy demand from the EIA Annual Energy Outlook 2020–Reference case[3], the future revenue that will be generated by the crude oil taxation policy over the next 4 years can be estimated. As a result, a total of $ 147.882 billion revenue is generated for every $5 per barrel increase in the crude oil tax over the next four years.

**Policy case study 2: renewable energy**. The electric power sector is heavily dominated (67%) by fossil fuels (coal, natural gas, petroleum, and other gases), whereas nuclear and renewable energy sources contribute about 17% and 16% of the remaining electricity generation, respectively[10] (see Supplementary Table 6). As a result, the electric power sector emits about 31.5% of the total US energy-related $CO_2$ emissions[69]. Thus, coordinated efforts for new policies and technologies are required so as to lessen the dependence on fossil fuels and subsequently reduce $CO_2$ emissions. To accomplish such reduction, the share of renewable energy within the electric power sector should be increased. This can be achieved either by setting a target renewable energy share for each power feedstock (analogous to the State based Renewable Portfolio Standards (RPS)) and/or by providing subsidies to the renewable energy generation (analogous to the Public Benefits Funds for Renewable Energy).

As of 2020, 30 US states, Washington D.C., and 3 US territories have adopted an RPS, whereas 7 US states and 1 US territory have set renewable energy goals for electricity generation[76]. The National Renewable Energy Laboratory indicates that these standards are most successful drivers of renewable energy projects when combined with tax credits[77]. However, the impact of these standards on the ratepayer are not clear and should be carefully

**Table 2 Average monthly difference ($/MMBtu), percentage increase (%), and revenue generated ($ billion) from January 2003 to June 2020.**

| Crude oil | Average monthly EPIC difference | Average monthly EPIC percentage increase | Total revenue | Average annual revenue |
|---|---|---|---|---|
| ($/barrel) | ($/MMBtu) | (%) | ($ billion) | ($ billion) |
| 2.5 | 0.248 | 1.37% | 302.886 | 17.308 |
| 5.0 | 0.497 | 2.73% | 605.772 | 34.616 |
| 7.5 | 0.745 | 4.10% | 908.658 | 51.923 |
| 10.0 | 0.994 | 5.46% | 1211.545 | 69.231 |
| 10.25 | 1.019 | 5.60% | 1241.833 | 70.962 |
| 12.5 | 1.242 | 6.83% | 1514.431 | 86.539 |
| 15.0 | 1.491 | 8.19% | 1817.317 | 103.847 |
| 17.5 | 1.739 | 9.56% | 2120.203 | 121.154 |
| 20.0 | 1.987 | 10.92% | 2423.089 | 138.462 |
| 22.5 | 2.236 | 12.29% | 2725.975 | 155.770. |
| 25.0 | 2.484 | 13.66% | 3028.862 | 173.078 |

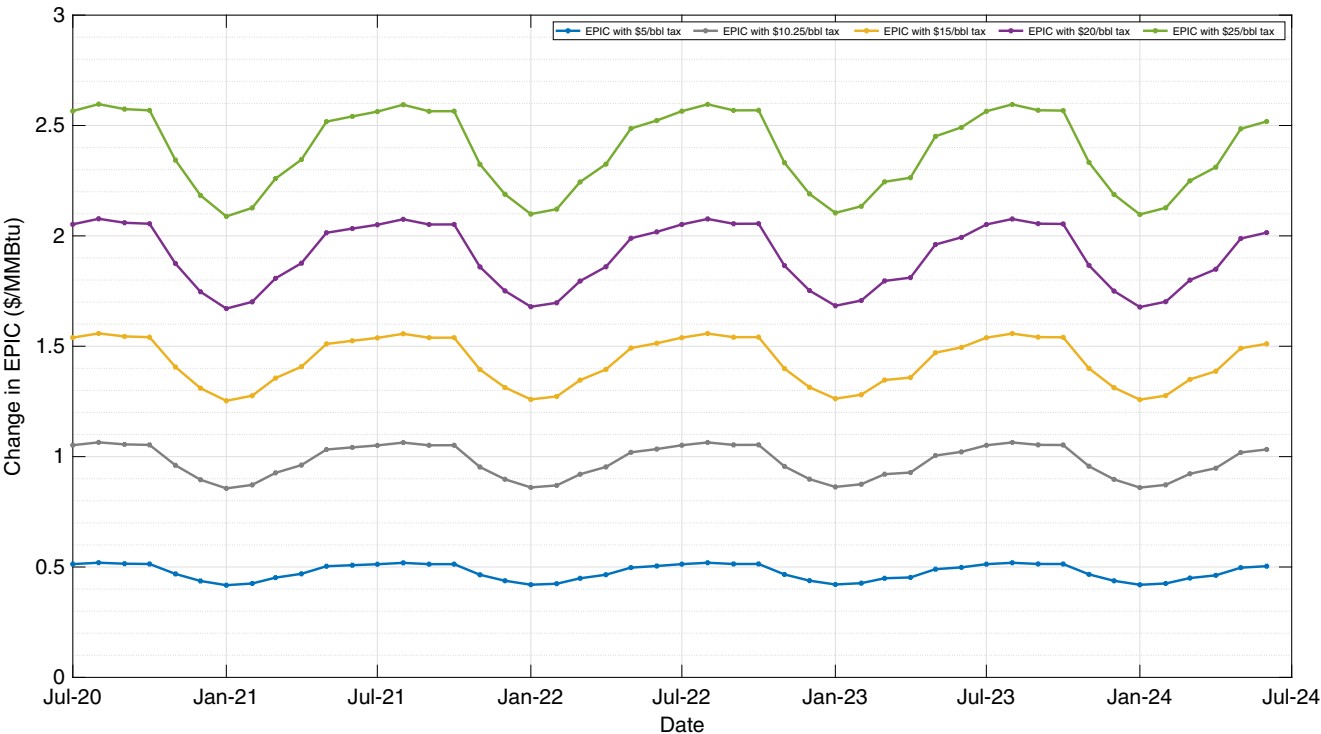

**Fig. 4 Change in EPIC with parametric crude oil tax over the next 4 years (July 2020 to June 2024).** A $10.25 per barrel of crude oil tax raises EPIC on average by $0.977/MMBtu over this 4-year period. A $25 per barrel of crude oil tax surges EPIC on average by $2.384/MMBtu in the same period. Source data are provided as a Source Data file.

**Table 3 Investigated weights for the non-fossil fuel feedstocks within the electric power sector.**

| Feedstock | Minimum weight (%) | Increment increase (%) | Maximum weight (%) |
|---|---|---|---|
| Nuclear | 18.0 | 3.0 | 30.0 |
| Hydroelectric power | 8.0 | 2.0 | 16.0 |
| Wind | 5.0 | 2.0 | 13.0 |
| Biomass | 0.5 | 0.25 | 1.5 |
| Solar | 1.0 | 1.0 | 5.0 |
| Geothermal | 0.3 | 0.1 | 0.7 |

evaluated. Although some reports claim that the benefits outweigh the costs of these standards[78,79], EPIC is an excellent tool to quantitatively analyze the costs of different renewable standards to the government and to the end-use consumers.

Therefore, in this policy case study, six non-fossil fuel feedstocks that are used in the electric power sector (nuclear, hydroelectric power, biomass, geothermal, solar, and wind) are investigated over a range of different target weights with tax credits/subsidies ranging from 0 to $9/MMBtu. The main assumptions, as well as the details for the calculations, are provided in the "Methods" section. Table 3 illustrates the grid of investigated target weights for each of the non-fossil feedstocks based on their nominal weights within the electric power sector.

Table 4 summarizes the results of this policy case study in terms of percentage change in EPIC at the maximum weight target for each non-fossil fuel feedstock in the past period (January 2003 to June 2020). It can be observed that nuclear energy causes a minor increase to EPIC at no tax credit, but as the tax credit increases, EPIC decreases significantly, for a maximum decline of −2.549%

corresponding to a tax credit of $9/MMBtu. Also, solar energy requires a subsidy of at least $6/MMBtu in order to lower the value of EPIC. It is also worth noting that increases either in the weights or in the tax credits of wind, hydroelectric, biomass, and geothermal energy always lead to a reduction in EPIC. This is also true even without a tax credit. For example, wind energy decreases EPIC from 0.177% up to 0.929% as the weight target increases with no tax credit, and from 0.621% up to 2.085% depending on the targeted weight with $9/MMBtu tax credit.

In Table 4, the average annual budget ($ million) required to provide subsidies at the maximum weight target for each non-fossil fuel feedstock in the same period (January 2003 to June 2020) is also presented. Clearly, the target weight and the tax credit are the key factors, affecting the annual budget. As either the target weight of each feedstock or the tax credit rises, the annual budget required to provide the relevant subsidy rises. Nuclear energy requires the highest subsidy budget (due to its maximum weight of 30%), but the corresponding decline in EPIC is also substantial (−2.549%) at the maximum level of tax credit.

Taking advantage of the excellent predictive ability of EPIC, the previous analysis can be extended to the future. As such, Fig. 5 demonstrates the effect on EPIC of various levels of tax credit applied to wind energy, for various weight levels of wind. The results for the remaining non-fossil feedstocks at different target weights and tax credits are presented in Supplementary Figs. 5–9.

At the lower end of tax credit (0 or 1$/MMBtu), the weight of the wind energy needs to be at least 11% so as to decrease the EPIC value, whereas at the higher end of tax credit (8 or 9 $/MMBtu), the EPIC value decreases even when weight contribution of wind energy is minimum (5%). Interestingly, as the percentage weight of wind energy increases within the electric power sector, EPIC decreases as the levelized cost of wind energy is rather low. As an example, even without any tax credit, EPIC decreases by 0.143% when wind energy provides 13% of the

**Table 4 Average % change in the EPIC and average annual budget ($ million) at the maximum weight target from January 2003 to June 2020.**

| Tax credit | Nuclear | | Hydroelectric | | Biomass | | Geothermal | | Solar | | Wind | |
|---|---|---|---|---|---|---|---|---|---|---|---|---|
| | (0.30) | | (0.16) | | (0.015) | | (0.007) | | (0.05) | | (0.13) | |
| ($/MMBtu) | (%) | ($ mil) | (%) | ($ mil) | (%) | ($ mil) | (%) | ($ mil) | (%) | ($ mil) | (%) | ($ mil) |
| 0 | 0.118% | 0 | −0.602% | 0 | −0.026% | 0 | −0.032% | 0 | 0.257% | 0 | −0.929% | 0 |
| 1 | −0.178% | 3787 | −0.760% | 2020 | −0.041% | 189 | −0.039% | 88 | 0.208% | 631 | −1.058% | 1641 |
| 2 | −0.475% | 7573 | −0.918% | 4039 | −0.056% | 379 | −0.046% | 177 | 0.158% | 1262 | −1.186% | 3282 |
| 3 | −0.771% | 11,360 | −1.076% | 6059 | −0.071% | 568 | −0.053% | 265 | 0.109% | 1893 | −1.315% | 4923 |
| 4 | −1.067% | 15,147 | −1.234% | 8078 | −0.086% | 757 | −0.060% | 353 | 0.059% | 2524 | −1.443% | 6564 |
| 5 | −1.364% | 18,933 | −1.392% | 10,098 | −0.100% | 947 | −0.067% | 442 | 0.010% | 3156 | −1.572% | 8204 |
| 6 | −1.660% | 22,720 | −1.550% | 12,117 | −0.115% | 1136 | −0.073% | 530 | −0.039% | 3787 | −1.700% | 9845 |
| 7 | −1.956% | 26,507 | −1.708% | 14,137 | −0.130% | 1325 | −0.080% | 618 | −0.089% | 4418 | −1.828% | 11,486 |
| 8 | −2.253% | 30,293 | −1.866% | 16,156 | −0.145% | 1515 | −0.087% | 707 | −0.138% | 5049 | −1.957% | 13,127 |
| 9 | −2.549% | 34,080 | −2.024% | 18,176 | −0.160% | 1704 | −0.094% | 795 | −0.187% | 5680 | −2.085% | 14,768 |

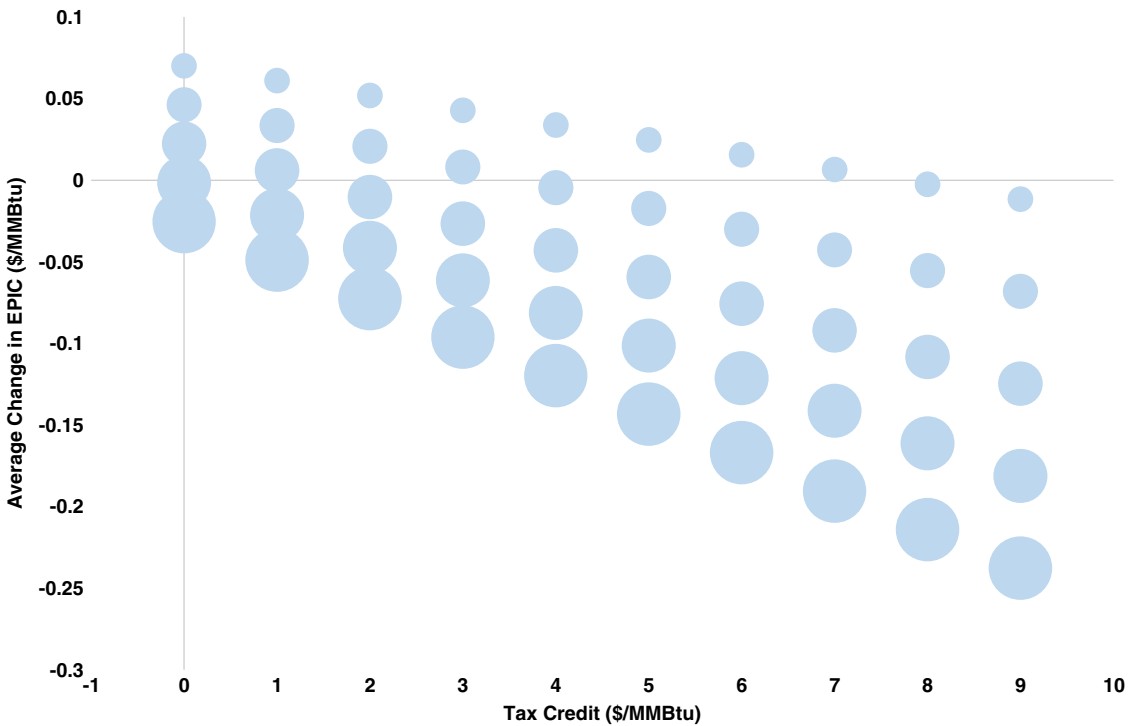

**Fig. 5 Wind power at different target weights (size of the bubble) and tax credits (x-axis), 2020–2024.** At maximum weight (13%) and without tax credit, EPIC decreases by 0.143% with no budget required, whereas at maximum weight (13%) and maximum tax credit ($9/MMBtu), EPIC decreases by 1.341% requiring around $16.5 billion annually from the government's budget. Source data are provided as a Source Data file.

electric power. Moreover, at the higher end of tax credit, the average decrease in EPIC exceeds $0.23/MMBtu.

Table 5 summarizes the average percentage change in EPIC and the average annual budget ($ million) required to provide the relevant subsidies in the future period from July 2020 to June 2024. The results are analogous with those from past period. More specifically, hydroelectric, wind, solar and geothermal power cause a drop of 0.198%, 0.143%, 0.090%, and 0.019%, respectively, in EPIC prices even with no tax credit. On the contrary, nuclear and biomass require a tax credit of at least $3/MMBtu and $4/MMBtu, respectively, so as reduce the value of EPIC. The most significant declines in EPIC are associated with potential subsidies of $9/MMBtu in the nuclear, hydroelectric, and wind power, resulting in a decline of 2.107%, 1.672%, and 1.341%, respectively. Nuclear energy is again expected to need the highest budget to provide the required subsidy, due to its maximum weight of 30%.

What is the price of energy to the end-users in the United States? The response is the EPIC. In this study, we introduce EPIC as the benchmark to calculate the average price of energy to the end-use consumers in the United States, in units of dollars per million Btu ($/MMBtu).

Can we estimate the impact of different policies in energy to the end-users? Yes, by using EPIC, a comprehensive, reliable, and easily interpretable instrument for policymakers to determine the quantitative effects of various policies.

EPIC is a novel predictive framework and its unique characteristics have been presented here. Its excellent predictive ability that enables the users to accurately determine the quantitative effects of these policies on the overall price of energy in the future has been also demonstrated. Two key policy case studies have been illustrated: the effects of a crude oil tax, and the implementation of subsidies for renewable energy. The scenarios have been investigated parametrically and the total revenue and budget/

**Table 5 Average % change in the EPIC and Average Annual Budget ($ million) at the maximum weight target from July 2020 to June 2024.**

| Tax credit | Nuclear | | Hydroelectric | | Biomass | | Geothermal | | Solar | | Wind | |
|---|---|---|---|---|---|---|---|---|---|---|---|---|
| | (0.30) | | (0.16) | | (0.015) | | (0.007) | | (0.05) | | (0.13) | |
| ($/MMBtu) | (%) | ($ mil) | (%) | ($ mil) | (%) | ($ mil) | (%) | ($ mil) | (%) | ($ mil) | (%) | ($ mil) |
| 0 | 0.657% | 0 | −0.198% | 0 | 0.054% | 0 | −0.019% | 0 | −0.090% | 0 | −0.143% | 0 |
| 1 | 0.350% | 4223 | −0.362% | 2252 | 0.039% | 211 | −0.026% | 99 | −0.141% | 704 | −0.276% | 1830 |
| 2 | 0.043% | 8445 | −0.525% | 4504 | 0.023% | 422 | −0.033% | 197 | −0.192% | 1408 | −0.409% | 3660 |
| 3 | −0.265% | 12,668 | −0.689% | 6756 | 0.008% | 633 | −0.040% | 296 | −0.244% | 2111 | −0.542% | 5489 |
| 4 | −0.572% | 16,891 | −0.853% | 9008 | −0.007% | 845 | −0.047% | 394 | −0.295% | 2815 | −0.675% | 7319 |
| 5 | −0.879% | 21,113 | −1.017% | 11,260 | −0.023% | 1056 | −0.055% | 493 | −0.346% | 3519 | −0.808% | 9149 |
| 6 | −1.186% | 25,336 | −1.181% | 13,513 | −0.038% | 1267 | −0.062% | 591 | −0.397% | 4223 | −0.941% | 10,979 |
| 7 | −1.493% | 29,559 | −1.344% | 15,765 | −0.053% | 1478 | −0.069% | 690 | −0.448% | 4926 | −1.075% | 12,809 |
| 8 | −1.800% | 33,781 | −1.508% | 18,017 | −0.069% | 1689 | −0.076% | 788 | −0.500% | 5630 | −1.208% | 14,639 |
| 9 | −2.107% | 38,004 | −1.672% | 20,269 | −0.084% | 1900 | −0.083% | 887 | −0.551% | 6334 | −1.341% | 16,468 |

subsidies required have been calculated. An increase of $10.25 per barrel in crude oil tax would have burdened the average energy related expenses per household by 5.54% in 2015, whereas $10.25 per barrel of crude oil tax will raise EPIC over the next 4 years by $0.997/MMBtu on average and will generate more than $300 billion over the same period. Increasing the percentage share of nuclear and renewable energy in the electric power sector will also assist towards tackling policy issues related to climate change. Moreover, the design of the policy case study using the EPIC framework has proved the unique ability of EPIC to determine the trade-offs among different energy sources within the electric power sector. Our results have shown that hydroelectric, wind, solar, and geothermal power will cause a drop in energy prices even with no tax credit. Hydroelectric and wind power should be the main areas of interest due to their higher impact in reducing the cost of electrical energy without requiring any subsidies.

For this study, the complete energy landscape of the Unites States, the four end-use sectors and the intermediate electric power sector have been thoroughly analyzed so as to identify the energy demand and the relevant prices of the energy products that serve as the backbone of EPIC. The introduction of a rolling horizon methodology has enabled us to overcome the problem associated with the three months lag of data availability for the weights of the feedstocks. This methodology also provides the necessary tools to accurately estimate these weights in the future. The predictive ability of this framework has been validated over a period of 174 months revealing a considerably low error between the actual values and the predictive values of the weights.

Future work can address different policy questions, such as how EPIC would respond to financial or monetary shocks, or to technological advancements. To address such questions, a simultaneous evaluation of multiple energy sources within the same framework across different production targets and subsidies would be implemented, taking into consideration potential limitations on the weights of each renewable energy source. Moreover, modeling of the up-to-date technological solutions for accurate representation of the levelized cost would be incorporated. Artificial intelligence (AI) methods could also be developed and implemented in different aspects of this framework such as forecasting the future prices of energy products for various forecasting horizons. The goal remains EPIC to be used for the design and optimization of a federal renewable energy policy for mitigating climate change, while ensuring that the price of energy remains affordable so as to not have a negative impact on short-term economic activity.

## Methods

**Rolling horizon methodology**. The proposed rolling horizon methodology uses data from the three previous periods in order to predict the data of interest in the current and future stages. The lookback period and the parameter estimation were selected considering the forecasting errors for different schemes and lookback periods.

Energy demand is highly seasonal[69] (see Supplementary Fig. 3 and Supplementary Table 5), so each month needs to be trained separately. Thus, all the schemes involve the parameter estimation for each month individually, so as to capture the seasonality effects that are crucial for accurate forecasting. Four different approaches for minimizing the sum of the squared error were tested over four different lookback periods (24, 36, 48, and 60 months), with each month being trained separately. Approaches 1 and 2 are "weight-based," whereas approaches 3 and 4 are "demand-based." The mathematical formulations are shown below for the lookback period of 36 months. Table 6 summarizes the results for the four approaches and four lookback periods.

Approach 1 - (weight-based)

$$\min \sum_m \text{Err}_m$$

$$\text{Err}_m = \sum_{m'} \left( \text{EPIC}_{m'} - \widehat{\text{EPIC}}_{m'} \right)^2$$

$$\widehat{\text{EPIC}}_{m'} = \sum_p (C_{m',p} * \widehat{w}_{m,p})$$

$$\text{EPIC}_m = \sum_p (C_{m,p} * w_{m,p})$$

$$\sum_p \widehat{w}_{m,p} = 1$$

$$\widehat{w}_{m,p} \geq 0$$

$$\forall m' | (m' - m) = (-36) \quad or \quad (-24) \quad or \quad (-12)$$

Approach 2 - (weight-based)

$$\min \sum_m \text{Err}_m$$

$$\text{Err}_m = \sum_{m',p} \left( w_{m',p} - \widehat{w}_{m,p} \right)^2$$

$$\sum_p \widehat{w}_{m,p} = 1$$

$$\widehat{w}_{m,p} \geq 0$$

$$\forall m' | (m' - m) = (-36) \quad or \quad (-24) \quad or \quad (-12)$$

where $\widehat{w}_{m,p}$ represents the predicted weight of product $p$ in month $m$.

Approach 3 - (demand-based)

$$\min \sum_m \text{Err}_m$$

$$\text{Err}_m = \sum_{m',p} \left( D_{m',p} - a_{m,p} * m' + b_{m,p} \right)^2$$

$$a_{m,p} * m + b_{m,p} \geq 0$$

$$\widehat{w}_{m,p} = \frac{a_{m,p} * m + b_{m,p}}{\sum_{p'} a_{m,p} * m + b_{m,p}}$$

$$\forall m' | (m' - m) = (-36) \quad or \quad (-24) \quad or \quad (-12)$$

**Table 6 Results on prediction of weights (different lookback periods).**

| | lookback_24 | | | | lookback_36 | | | |
|---|---|---|---|---|---|---|---|---|
| | **App1** | **App2** | **App3** | **App4** | **App1** | **App2** | **App3** | **App4** |
| Min error | 0.052% | 0.049% | 0.064% | 0.067% | 0.058% | 0.053% | 0.051% | 0.127% |
| Max error | 0.263% | 0.221% | 0.410% | 0.351% | 0.292% | 0.228% | 0.290% | 0.624% |
| Average error | 0.106% | 0.099% | 0.147% | 0.154% | 0.121% | 0.104% | 0.127% | 0.336% |
| | lookback_48 | | | | lookback_60 | | | |
| | **App1** | **App2** | **App3** | **App4** | **App1** | **App2** | **App3** | **App4** |
| Min error | 0.064% | 0.057% | 0.052% | 0.233% | 0.064% | 0.065% | 0.054% | 0.394% |
| Max error | 0.265% | 0.209% | 0.289% | 0.776% | 0.261% | 0.196% | 0.255% | 0.839% |
| Average error | 0.137% | 0.107% | 0.120% | 0.454% | 0.158% | 0.113% | 0.115% | 0.536% |

Approach 4 - (demand-based)

$$\min \sum_m \text{Err}_m$$

$$\text{Err}_m = \sum_p \left( D_{m-12,p} - (a_{m,p} * D_{m-24,p} + b_{m,p} * D_{m-36,p}) \right)^2$$

$$a_{m,p} * D_{m-12,p} + b_{m,p} * D_{m-24,p} \geq 0$$

$$\widehat{w}_{m,p} = \frac{a_{m,p} * D_{m-12,p} + b_{m,p} * D_{m-24,p}}{\sum_{p'} a_{m,p'} * D_{m-12,p'} + b_{m,p'} * D_{m-24,p'}}$$

where $D_{m,p}$ represents the demand of product $p$ in month $m$, and $a_{m,p}$ and $b_{m,p}$ represent the fitted parameter 1 and 2 of product $p$ in month $m$, respectively.

As it can be seen, approaches 1 and 2 outperform approaches 3 and 4, regardless of the lookback period. Between the weight-based approaches, approach 2 is the one with the lowest errors regardless of the lookback period, so it is the one selected. With regards to the lookback periods, the cases for 24 and 36 months produce the lowest errors in comparison to the 48 and 60 months. As the results for the 24 and 36 months are comparable, the longer lookback period is selected, which will capture better the increasing volatility of the energy demand in the future. Therefore, the methodology selected is Approach 2 with 36 months lookback period.

**EPIC framework.** The two key factors comprising EPIC are the total demand of the energy products that are directed to the end-use sectors in the United States along with their respective prices.

The sectors of the US economy consume products originating from crude oil, natural gas, coal, nuclear, solar, wind, hydroelectric, geothermal, and several types of biomass. The exact determination of these products, their consumption levels, and their monthly prices is critical to the construction of EPIC. The full list of these products and the corresponding sector they are consumed in are presented in the Supplementary Table 2. The monthly consumption (in energy units) along with the monthly price (in $ per energy unit) for each of the energy products is extracted from the data provided by the EIA and other sources, and are shown in the Supplementary Table 4.

The real weight fraction based on the demand of each of the selected 56 energy products is calculated using Equation 1:

$$w_{m,p} = \frac{D_{m,p}}{\sum_p D_{m,p}} \qquad \forall (m,p) \tag{1}$$

where $w_{m,p}$ is the weight fraction of product $p$ in month $m$ and $D_{m,p}$ is the demand of product $p$ in month $m$.

EPIC is formulated as the summation of the price (in $/MMBtu) of each product multiplied by the weight fraction of each product. The unit of EPIC is $/MMBtu. EPIC represents the average price of energy in a given month and its formulation is presented in Equation 2:

$$\text{EPIC}_m = \sum_p w_{m,p} * C_{m,p} \qquad \forall m \tag{2}$$

where $\text{EPIC}_m$ represents the value of EPIC in month $m$, $C_{m,p}$ represents the price of product $p$ in month $m$, and $w_{m,p}$ is the weight fraction of product $p$ in month $m$.

Due to the time lag in data availability and the need for future forecasts for our policy applications as well, a rolling horizon based model is developed to predict the weights of the demand of each energy product at a present or future stage of interest using the data from the previous 3 years. It is worth mentioning that each month is trained separately since the energy demand is highly seasonal[69] (see Supplementary Fig. 3 and Supplementary Table 5).

The weights of the energy products are predicted using a weight-based objective function, which minimizes the squared difference between the real value of a product's weight in the past horizon and the predicted value of the product's weight for the month of interest. The optimization model takes into account the

data from the previous 3 years and is stated as:

$$\min \sum_m \text{Err}_m$$

$$\text{Err}_m = \sum_{m',p} \left( w_{m',p} - \widehat{w}_{m,p} \right)^2$$

$$\sum_p \widehat{w}_{m,p} = 1 \tag{3}$$

$$\widehat{w}_{m,p} \geq 0$$

$$\forall m' | (m' - m) = (-36) \quad \text{or} \quad (-24) \quad \text{or} \quad (-12)$$

where $\widehat{w}_{m,p}$ represents the predicted weight of product $p$ in month $m$.

The predictive ability of this framework is assessed using the squared prediction error for each month over a period of 174 months from January 2006 to June 2020 and the results are presented in the relevant section. The following formula is used:

$$\text{PredErr}_m = \sum_p \left( w_{m,p} - \widehat{w}_{m,p} \right)^2 \tag{4}$$

**Policy case study 1: crude oil.** The main assumptions for this policy case study are:

- Crude oil has a heating content of 5.721 MMBtu per barrel[69].
- The petroleum refinery efficiency is 90%.
- The amount of petroleum and petroleum products being sent for electricity generation is assumed to be negligible ($\approx 0.57\%$ over the last year).
- Crude oil demand is considered as inelastic in the short run, with long-run values of elasticity being generally higher in absolute values but still well below $1$[71–74]. As crude oil is inelastic, the investigated crude oil tax has not affected the historical values of crude oil consumption.
- The future effects (up to 2024) are assessed using the predicted values for the weights of the petroleum products applying the methodology that is described in the relevant section.
- The future annual demand as well as the future nominal weights within the petroleum sector are estimated using data from the EIA Annual Energy Outlook 2020–Reference case[3].

The revised EPIC from the subject policy case study is estimated for the past and for the future as follows:

$$\text{Past}: \text{EPIC}_{\text{policy}} = \text{EPIC}_{\text{without}_{\text{Tax}}} + \text{Tax}_{\text{crude}} * \frac{1}{\text{HeatContent}_{\text{crude}}} * \frac{1}{\text{RefEff}_{\text{crude}}} *$$
$$\sum_{p \in P \backslash '\text{petprod}'} w_{m,'\text{petprod}'} \tag{5}$$

$$\text{Future}: \Delta\text{EPIC}_{\text{policy}} = \text{Tax}_{\text{crude}} * \frac{1}{\text{HeatContent}_{\text{crude}}} * \frac{1}{\text{RefEff}_{\text{crude}}} *$$
$$\sum_{p \in P \backslash '\text{petprod}'} w_{m,'\text{petprod}'} \tag{6}$$

where $\text{Tax}_{\text{crude}}$ stands for the parametric crude oil tax in $/barrel, $\text{HeatContent}_{\text{crude}}$ stands for the heating content of crude oil, $\text{RefEff}_{\text{crude}}$ stands for the petroleum refinery efficiency, and $w_{m,'\text{petprod}'}$ stands for the weight of petroleum products "petprod" in month $m$.

**Policy case study 2: renewable energy.** The main assumptions for this policy case study are:

- The effect of the policy is investigated independently for each feedstock.
- The production targets/subsidies affect only the electric power sector, so only the relative weights within the electric power sector change.

- The target weights are attainable with the existing resources and at the current production costs.
- When a specific target weight is enforced on an electricity energy feedstock, the remaining feedstock weights are normalized to add up to 1.
- The levelized cost of the energy feedstocks is taken from Lazard's Levelized Cost of Energy Analysis report for the period 2008 to 2013[80–85] and the EIA Annual Energy Outlook for period 2014 to 2020[86–92], with the exception of data for the petroleum liquids, which are also taken from Lazard's Levelized Cost of Energy Analysis reports. The data from 2008 are used for the period from 2003 to 2007 (Supplementary Table 6).
- The future effects (up to 2024) are assessed using the predicted values for the weights of the electricity products applying the methodology that is described in the relevant section.
- The future annual demand and the future nominal weights within the electric power sector are estimated using data from the EIA Annual Energy Outlook 2020–Reference case[3].

Once a target weight for a renewable feedstock has been set, all other weights within the electric power sector are re-normalized as follows:

$$w_{norm,f} = w_f^{old} * \frac{1 - w_{f'}^{target}}{1 - w_{f'}^{old}} \qquad (7)$$

where $f'$ represents the feedstock investigated and f represents all other feedstocks.

The change in EPIC due to the new target weight is then calculated as follows:

$$\Delta EPIC_1 = w_{elec} * \sum_f Cost_f * (w_{norm,f} - w_f^{old}) + Cost_{f'} * (w_{f'}^{target} - w_{f'}^{old}) \qquad (8)$$

where $w_{elec}$ stands for the aggregate weight of electricity (i.e., products 50–53), whereas $Cost_{f'}$ and $Cost_f$ stand for the levelized cost of electricity production from feedstock $f'$ and f, respectively. When this delta term becomes positive, meaning that the value of EPIC increases, the cost of the targeted feedstock is higher than the average cost of the displaced feedstocks. On the contrary, when this delta term becomes negative, the cost of the targeted feedstock is lower than the average cost of the displaced feedstocks and so the value of EPIC decreases.

The change in EPIC due to subsidies is calculated as follows:

$$\Delta EPIC_2 = w_{elec} * w_{f'}^{target} * Tax_{cred} \qquad (9)$$

where $Tax_{cred}$ represents the subsidy in $/MMBtu.

The revised EPIC from the subject policy case study is estimated for the past as well as for the future as follows:

$$Past : EPIC_{policy} = EPIC + \Delta EPIC_1 + \Delta EPIC_2 \qquad (10)$$

$$Future : \Delta EPIC_{policy} = \Delta EPIC_1 + \Delta EPIC_2 \qquad (11)$$

**Reporting summary**. Further information on research design is available in the Nature Research Reporting Summary linked to this article.

## Data availability

Figures 1 and 3, and Supplementary Figs. 1 and 2 do not have associated data. All data used for this analysis are available from cited publicly available sources or from the corresponding author upon reasonable request. Source data are provided with this paper.

## Code availability

The optimization code in GAMS that supports the analysis within this paper and other findings of this study are available from the corresponding author upon reasonable request.

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

## Acknowledgements
We express our gratitude for the financial support from the Texas A&M Energy Institute and the Mays Business School.

## Author contributions
All authors worked in the methodology and the development and analysis of the case studies. S.G.B., A.M.Z., O.O., and L.R.M. contributed to the writing of the paper, while C.A.F., D.R.H., S.M.S., and E.N.P. supervised the project and edited the paper.

## Competing interests
The authors declare no competing interests.
