## [Peer Review File · Nature Communications]

REVIEWER COMMENTS

Reviewer #1 (Remarks to the Author):

The study by Stefanos G. Baratsas et al. presents a method to predict end-use energy prices. Although the paper is well written and makes an excellent sales pitch, it does not meet the standards of a journal such as Nature Communications. I strongly recommend rejecting this paper for several reasons.

First, the paper claims the method to be novel without showing how it compares against other methods in the literature. Second, basically the EPIC is a price index calculated as the weighted average of prices. How is that novel? Energy analysts have been doing this for ages. Third, the section leading upto Figure 1 is so basic it felt as if I was reading a very basic primer on energy systems. Figure 1 is extremely basic information. It is directly copied from EIA.

Fourth, the authors do not present the data used clearly. What's the spatial scope and resolution of the data? Can this method be used to predict prices, say, at the state-level? Fifth, to what degree can this method be extended to other countries? Sixth, what kinds of new data analytic tools such as AI methods have been used in this study?

Reviewer #2 (Remarks to the Author):

The authors claim that they have developed a novel framework, an index to calculate the average price of energy in the United States. They determine the total demand of the energy products directed to the end-use sectors, and the corresponding price of each product. The claim to determine the future electricity prices through rolling horizon methodology is novel. This manuscript will add value in the existing literature of energy.

However, I have minor concerns, as below:

1- As per figure 1, US energy mix constitutes around 80% on non-renewable sources. So, in such context I recommend that authors have to apply EPIC on this domain too.

2- Besides, I have not find the reason to include just three lag periods to predict the data of interest in current stage.

Reviewer #1 (Remarks to the Author):

The study by Stefanos G. Baratsas et al. presents a method to predict end-use energy prices. Although the paper is well written and makes an excellent sales pitch, it does not meet the standards of a journal such as Nature Communications. I strongly recommend rejecting this paper for several reasons.

First, the paper claims the method to be novel without showing how it compares against other methods in the literature. Second, basically the EPIC is a price index calculated as the weighted average of prices. How is that novel? Energy analysts have been doing this for ages. Third, the section leading up to Figure 1 is so basic it felt as if I was reading a very basic primer on energy systems. Figure 1 is extremely basic information. It is directly copied from EIA.

Fourth, the authors do not present the data used clearly. What's the spatial scope and resolution of the data? Can this method be used to predict prices, say, at the state-level? Fifth, to what degree can this method be extended to other countries? Sixth, what kinds of new data analytic tools such as AI methods have been used in this study?

Response: We would like to thank the first reviewer for the interest in our work and the constructive feedback. Please find the responses and the corresponding actions for each of your comments below.

Comment R1.1: *First, the paper claims the method to be novel without showing how it compares against other methods in the literature.*

Response R1.1: Please refer to **Action 1** on the list of Actions for the corrections.

The Introduction section has been expanded providing evidence of the novelty of this work.

Comment R1.2: *Second, basically the EPIC is a price index calculated as the weighted average of prices. How is that novel? Energy analysts have been doing this for ages.*

Response R1.2: Please refer to **Action 2** on the list of Actions for the corrections.

Following our response on the previous comment (R1.1), another paragraph has been added in the Introduction section for a comparison of the proposed index against the most widely used indices in the energy sector.

Comment R1.3: *Third, the section leading up to Figure 1 is so basic it felt as if I was reading a very basic primer on energy systems. Figure 1 is extremely basic information. It is directly copied from EIA.*

Response R1.3: Please refer to **Action 3** on the list of Actions for the corrections.

We used a whole section for the US energy landscape, because it is a complex and extensive network of different energy feedstocks and products that can be used in different ways before reaching the end users. We wanted to explicitly define the terms “demand”, and “products” because these are crucial for our analysis. Also, we wanted to emphasize the importance of maintaining a holistic and precise methodology throughout this context in order to avoid any miscalculation as well as to assist others who may want to reproduce this work. To this respect, we decided to adapt the terminology from EIA which gives us access to a vast amount of information, while at the same time it makes it easier for our readers to follow up this work. Finally, we consider essential a figure that illustrates the complexity of the energy landscape, assisting especially the non-technical audience to clearly visualize it.

Comment R1.4: *Fourth, the authors do not present the data used clearly. What's the spatial scope and resolution of the data? Can this method be used to predict prices, say, at the state-level?*

Response R1.4: Please refer to **Action 4** on the list of Actions for the corrections.

An excel file with the source data for the figures 2, 4, 5 and the Supplementary Figures is provided. Supplementary Table 2 and 3 have been updated with detailed references for the data sources for both the demand and the prices of the energy products. Also, two new sections have been added in the Supplementary Notes (2 and 3) that provide step by step details of the preliminary calculations for the preparation of the data to be used in the model. The proposed methodology can be easily extended to state and regional levels, provided that a thorough analysis of the specific energy landscape has been conducted, the energy feedstocks and products have been identified, and data for their prices and demands are available.

Comment R1.5: *Fifth, to what degree can this method be extended to other countries?*

Response R1.5: Please refer to **Action 5** on the list of Actions for the corrections.

Following our response on the previous comment (R1.4), this methodology can be extended to other countries, provided that a thorough analysis of the specific energy landscape has been conducted, the energy feedstocks and products have been identified, and data for their prices and demands are available.

Comment R1.6: *Sixth, what kinds of new data analytic tools such as AI methods have been used in this study?*

Response R1.6: Please refer to **Action 6** on the list of Actions for the corrections.

In this work, we have developed and presented an optimization based rolling horizon model. The model is written and executed in GAMS 31.2.0, and the data preparation is conducted in Microsoft Excel.

In the presented methodology, we have not used any AI methods. However, AI methods can be developed and implemented as future work in different aspects of this framework such as forecasting the future prices of energy products for various forecasting horizons using Neural Networks and other advanced forecasting methods.

Reviewer #2 (Remarks to the Author):

The authors claim that they have developed a novel framework, an index to calculate the average price of energy in the United States. They determine the total demand of the energy products directed to the end-use sectors, and the corresponding price of each product. The claim to determine the future electricity prices through rolling horizon methodology is novel. This manuscript will add value in the existing literature of energy.

However, I have minor concerns, as below:

1- As per figure 1, US energy mix constitutes around 80% on non-renewable sources. So, in such context I recommend that authors have to apply EPIC on this domain too.

2- Besides, I have not found the reason to include just three lag periods to predict the data of interest in current stage.

Response: We would like to thank the second reviewer for the interest in our work and the kind and constructive feedback. Please find the responses and the corresponding actions for each of your comments below.

Comment R2.1: *As per figure 1, US energy mix constitutes around 80% on non-renewable sources. So, in such context I recommend that authors have to apply EPIC on this domain too.*

Response R2.1: Please refer to **Action 7** on the list of Actions for the corrections.

We have added another policy case study for a potential tax in crude oil. We selected crude oil because it represents the most dominant from the non-renewable sources.

Comment R2.2: *Besides, I have not found the reason to include just three lag periods to predict the data of interest in current stage.*

Response R2.2: Please refer to **Action 8** on the list of Actions for the corrections.

Four different parameter estimation schemes and four different lookback periods were evaluated, and the final results along with our justification are shown.

List of Actions for the Revised Manuscript:

What is the price of energy to the end-users? A novel predictive framework with ~~and one of its~~ applications to the energy and monetary policies

Action 1 – The following paragraph is added in the “Introduction” section:

“The novelty of the proposed index (EPIC) lies on its unique features. Firstly, it represents the average price of energy in the US over the entire energy landscape covering all the different energy sources and feedstocks (non-renewables and renewables) as well as the end-use sectors. As such, it is not just a representative price of a sub-section of the energy landscape such as the price of electricity in the residential sector or the price of oil products in the industrial sector which is common in the existing literature [1-4]. Secondly, the proposed formulation collectively captures the two key attributes of energy, the supply and demand mechanisms along with the prices of the energy feedstocks and products across the entire energy landscape. This is another unique feature since the methodologies in the literature generally focus on specific energy sectors [5-7]. Thirdly, the excellent forecasting ability of the proposed mathematical framework allows the estimation of the current value of EPIC, thus the current price of energy, overcoming the issue of the non-availability of actual data while it can also be used to forecast future values of the energy demand accurately. In contrast, the forecasting frameworks in the literature focus on specific energy sectors [8] such as electricity [9-11], natural gas [12], crude oil [13] and petroleum products [14], with generally much shorter forecasting horizon [8-14]. Finally, it is a quantitative approach to evaluate, design and optimize different policy questions of significant public interest such as a policy case study for the renewable energy, and a policy case study for the crude oil tax.”

[1] U.S. Energy Information Administration. Electric Power Monthly with Data for September 2019. Tech. Rep., Washington, DC, USA (November 2019).

[2] Asche, F., Gjølberg, O. & Völker, T. Price relationships in the petroleum market: an analysis of crude oil and refined product prices. *Energy Econ.* 25, 289–301 (2003).

[3] Bacon, R., Chadwick, M., Dargay, J., Long, D. & Mabro, R. Demand, prices and the refining industry: a case-study of the European oil products market (Oxford University Press, 1990).

[4] Lanza, A., Manera, M. & Giovannini, M. Modeling and forecasting cointegrated relationships among heavy oil and product prices. *Energy Econ.* 27, 831–848 (2005).

[5] Silvente, J., Kopanos, G. M., Pistikopoulos, E. N. & Espuña, A. A rolling horizon optimization framework for the simultaneous energy supply and demand planning in microgrids. *Appl. Energy* 155, 485–501 (2015).

[6] Kanamura, T. A supply and demand based volatility model for energy prices. *Energy Econ.* 31, 736–747 (2009).

[7] Faria, P. & Vale, Z. Demand response in electrical energy supply: An optimal real time pricing approach. *Energy* 36, 5374–5384 (2011).

[8] Suganthi, L. & Samuel, A. A. Energy models for demand forecasting—a review. *Renew. sustainable energy reviews* 16, 1223–1240 (2012).

[9] Taylor, J. W. Short-term electricity demand forecasting using double seasonal exponential smoothing. *J. Oper. Res. Soc.* 54, 799–805 (2003).

[10] Gonzalez-Romera, E., Jaramillo-Moran, M. A. & Carmona-Fernandez, D. Monthly electric energy demand forecasting based on trend extraction. *IEEE Transactions on power systems* 21, 1946–1953 (2006).

- [11] Hyndman, R. J. & Fan, S. Density forecasting for long-term peak electricity demand. *IEEE Transactions on Power Syst.* 25, 1142–1153 (2009).
- [12] Soldo, B. Forecasting natural gas consumption. *Appl. Energy* 92, 26–37 (2012).
- [13] Xiong, J. & Wu, P. An analysis of forecasting model of crude oil demand based on cointegration and vector error correction model (vec). In *2008 International Seminar on Business and Information Management*, vol. 1, 485–488 (IEEE, 2008).
- [14] Grushevenko, E., Mitrova, T., Kulagin, V., Grushevenko, D. & Galkina, A. Complex method of petroleum products demand forecasting considering economic, demographic and technological factors. *Econ. Bus. Lett.* 4, 98–107 (2015).

Action 2 – The following paragraph is added in the “Introduction” section:

“The EPIC framework demonstrates novel features not only for the existing academic literature but also for the financial tools in the energy sector. The various energy indices are primarily capitalization weighted indices, capped market capitalization indices, price weighted indices and world production weighted indices [15] with some of the most representative examples being S&P 500 Energy Index [16], MSCI US IMI Energy 25/50 Index [17], and S&P GSCI Energy [18]. S&P 500 Energy Index tracks the market capitalization of energy companies and their stock price, while MSCI US IMI Energy 25/50 Index captures the large, mid and small cap segments of the US equity universe, but neither of them reflects to the actual energy products. The S&P GSCI Energy includes only the futures contracts on physical commodities with the weights being calculated based on the world production of these commodities, however it focuses exclusively on oil and gas products, without capturing other energy feedstocks such as electricity, or renewables. On the contrary, EPIC captures the prices of all energy feedstocks while the weights are calculated from the actual demands of these energy feedstocks. More details about these financial indices, their constituents and their weights are shown in the Supplementary Table 1.”

[15] Dow Jones S&P Indices. S&P Index Mathematics Methodology. Tech. Rep. (June 2020).

[16] Dow Jones S&P Indices. S&P U.S. Indices Methodology. Tech. Rep. (May 2020).

[17] MSCI. MSCI U.S. IMI Energy 25/50 Index. Tech. Rep. (May 2020).

[18] Dow Jones S&P Indices. S&P GSCI Methodology. Tech. Rep. (May 2020).

Action 3 - Figure 1 has been replaced with a new representation of the US energy landscape. The relevant section has been amended as follows:

“Figure 1 illustrates in detail the complete U.S energy landscape for the different energy feedstocks by source and end-use sector. Moreover, the energy landscape of each energy feedstock is shown in Supplementary Figures 1a-1e.”

Figure 1: US energy landscape by source and end-use sector

Supplementary Information Material

Figure 1a: Landscape of Crude Oil in the United States

Figure 1b: Landscape of Natural Gas and Coal in the United States

Figure 1c: Landscape of Solar and Wind in the United States

Figure 1d: Landscape of Geothermal and Hydroelectric in the United States

Figure 1e: Landscape of Biomass in the United States

Action 4 – Supplementary Table 2 presents the 56 energy products and the corresponding sectors that are consumed in. The terminology has been selected so as to match the terminology of the main data sources i.e. EIA, Bureau of Labor Statistics, Lazard etc. Supplementary Table 3 contains detailed references for the data sources for the demand and prices of the energy products. With regards to the demand data, the particular tables from the EIA Monthly Energy Report and the EIA Electric Power Monthly Report are listed. Similarly, for the price data, the specific tables from the EIA Monthly Energy Report, EIA Annual Energy Outlook, EIA Electric Power Monthly Report and EIA State Energy Data System are listed when applicable, along with detailed references from the other data sources and databases i.e. Lazard Annual reports, DoE reports, Bureau of Labor Statistics and Thomson Reuters databases. All data used for this analysis are available from cited publicly available sources.

Furthermore, the source data underlying Figures 2, 4, 5 and the Supplementary Figures along with the data for the demand and the prices of all energy products are provided as a Source Data file.

We have also added two new sections, Supplementary Note 2 and 3, that contain details about the preliminary calculations for the preparation of demand and price data, which are used in the model. Specifically, for the price data, we provide step by step details for a few indicative products, since similar procedures are applied to the remaining ones. For researchers who are interested to see all analytical details or wish to reproduce our results, the full documentation will be provided in our websites (<https://energy.tamu.edu/>, <http://parametric.tamu.edu/>).

This methodology can be applied to state or regional levels, and can be extended even to other countries, provided that an analysis of the particular energy landscape has been conducted so as to identify all the applicable energy feedstocks and products, and that accurate data of the demands and prices of the energy components are available.

The following paragraph is added in the “Energy Price Index (EPIC) framework” section:

“Please note that the proposed framework is generic and can be applied to a) the U.S on a national level, b) to U.S on a state by state basis, c) regional level of multi-states, and d) other countries, provided that a thorough analysis of the specific energy landscape has been conducted, the particular energy feedstocks and products have been identified, and data for their prices and demands are available.”

The following sections have been added:

“Supplementary Note 2: Extra Details for the Demands of the Energy Products

- Demand data for Products 1 to 45 are provided in *Trillion BTUs* for each month.
- Natural Gas Consumption (Products 46-49) are provided in *Billion Cubic Feet* for each month. A conversion factor of 1,036 BTU per cubic foot is used to convert into units of energy. [19]
- Electricity Consumption (Products 50-53) are provided in *Thousand Megawatt hours* for each month. A conversion factor of 3,412 BTU per kWh is used to convert into units of energy. [20]
- Coal Consumption (Products 54-56) are provided in *Thousand Short Tons* for each month. For the residential and commercial coal consumption (Products 54-55), a conversion factor of 19.268 MMBTU per short ton is used to convert into units of energy, while for the industrial coal consumption (Product 56) a conversion factor of 28.608 MMBTU per short ton is used. [21]

[19] U.S. Energy Information Administration. Monthly Energy Review, Table A4. Tech. Rep., Washington, DC, USA (December 2019).

[20] U.S. Energy Information Administration. Monthly Energy Review, Table A6. Tech. Rep., Washington, DC, USA (December 2019).

[21] U.S. Energy Information Administration. Monthly Energy Review, Table A5. Tech. Rep., Washington, DC, USA (December 2019).

Supplementary Note 3: Extra Details for the Prices of the Energy Products

- 1) Distillate fuel oil consumed by the residential sector
 - a) The U.S. No. 2 Heating Oil Residential prices (Dollars per Gallon) are used. Prices are given for 6 months of the year (October – March). For the rest of the months, the No. 2 Heating Oil New York Harbor Spot Prices (\$/gal) are used to estimate the distillate fuel oil prices using linear regression.
 - b) A conversion factor of 138,490 BTU/gal is used.
 - c) Federal tax and average state tax are added on top of the abovementioned prices (<https://www.eia.gov/petroleum/marketing/monthly/xls/fueltaxes.xls>)
- 2) Kerosene consumed by the residential sector
 - a) The prices of U.S. Kerosene Retail Sales by Refiners (\$/gal) are used. If a price is not available for a month, then the prices of the U.S. Kerosene-Type Jet Fuel Retail Sales by Refiners are used to estimate the price of price using linear regression.
 - b) A conversion factor of 0.135 MMBTU/gal is used.
 - c) Federal tax and average state tax are added on top of the abovementioned prices. (<https://www.eia.gov/petroleum/marketing/monthly/xls/fueltaxes.xls>)
- 3) Hydrocarbon gas liquids (Propane) consumed by the residential sector
 - a) The U.S. Propane Residential prices (Dollars per Gallon) are used. Prices are given for 6 months of the year (October – March). For the rest of the months, the Mont Belvieu, TX Propane Spot Price FOB (\$/gal) are used to estimate the residential hydrocarbon gas liquid prices using linear regression.
 - b) A conversion factor of 0.09133 MMBTU/gal is used.

- c) Federal tax is added on top of the abovementioned prices (<https://afdc.energy.gov/fuels/laws/LPG?state=us>)
- 4) Distillate fuel oil consumed by the commercial sector
 - a) The U.S. No. 2 Fuel Oil Retail Sales by Refiners (Dollars per Gallon) are used
 - b) A conversion factor of 0.13849 MMBTU/gal is used.
 - c) Federal tax and average state tax are added on top of the above-mentioned prices. (<https://www.eia.gov/petroleum/marketing/monthly/xls/fueltaxes.xls>)
- 10) Asphalt and Road oil consumed by the industrial sector
 - a) The Asphalt and road oil average prices, for all end-use sectors in the United States (Dollars per MMBtu) are used from State Energy Data System (SEDS) [21]. Since the data are presented annually, their monthly values are estimated through a linear regression with the data of the Producer Price Index (PPI) of Asphalt (PPI by Industry: Petroleum Refineries: Asphalt).
 - b) No tax is added as per the SEDS.
- 19) Aviation gasoline consumed by the transportation sector
 - a) The U.S. Aviation Gasoline Retail Sales by Refiners (Dollars per Gallon) are used. In case there are no available data for a month, a linear regression of the retail sales and the U.S. Aviation Gasoline Wholesale/Resale Price by Refiners (Dollars per Gallon) is used to determine the retail price of that month.
 - b) A conversion factor of 0.12019 MMBTU/gal is used.
 - c) No tax is added as per the SEDS
- 26) Geothermal energy consumed by the residential sector
 - a) The levelized cost of energy (\$/MWh) from Lazard's Levelized Cost of Energy Analysis report is used. The average value between the low and the high values of geothermal that are provided in the report is used.
 - b) A conversion factor of 0.29308 MWh/MMBTU is used.
 - c) A constant value is taken for all months of the year.
- 47) Natural gas consumed by the commercial sector
 - a) The U.S. Price of Natural Gas Sold to Commercial Consumers (Dollars per Thousand Cubic Feet) is used.
 - b) A conversion factor of 1.036 Million Btu/Thousand cubic Feet is used.
 - c) Taxes are included in the price.
- 52) Electricity consumed by the industrial sector
 - a) The average retail price of electricity in the industrial sector (cents per kWh) is used.
 - b) A conversion factor of 0.003412 Million Btu/kWh is used
 - c) Taxes are included in the price.
- 55) Coal consumed by the commercial sector
 - a) The Coal price in the commercial sector (Dollar per MMBtu) is used. Since the data are presented annually, their monthly values are estimated through a linear regression with the data of the Producer Price Index (PPI) by Commodity for Fuels and Related Products and Power: Coal.
 - b) Taxes are included in the price

[21] U.S. Energy Information Administration. State Energy Data System 2017. Tech Rep., Washington, DC, USA (2017)."

Action 5 – The following paragraph is added in the “Energy Price Index (EPIC) framework” section: “Please note that the proposed framework is generic and can be applied to a) the U.S on a national level, b) to U.S on a state by state basis, c) regional level of multi-states, and d) other countries, provided that a thorough analysis of the specific energy landscape has been conducted, the particular energy feedstocks and products have been identified, and data for their prices and demands are available.”

Action 6 – The following paragraph is added in the “Conclusions” section: “AI methods could also be developed and implemented in different aspects of this framework such as forecasting the future prices of energy products for various forecasting horizons.”

Action 7 – A new policy case study for a new crude oil tax is added. The new section is entitled “Policy case study 1 - Crude oil”, and describes the effects on the price of energy from a tax in the crude oil parametrically, along with the revenue that would be generated by such a policy retrospectively and prospectively.

“Policy case study 1 - Crude Oil

The proposed framework can be used for a wide range of policy questions and analyses, and a potential tax in crude oil is investigated here as an alternative policy for mitigating climate change and concurrently generating substantial revenue that is required for climate finance. Such a policy of US\$10.25 per barrel tax on crude oil was also proposed by U.S President Barack Obama back in 2016 to support new transportation systems designed to reduce carbon emissions and congestion[24].

Here, we examine parametrically the effects of a crude oil tax ranging from US\$2.5 per barrel up to US\$25 per barrel in EPIC, and calculate the amount of revenue that could be generated by such a policy from January 2003 until September 2019. Taking advantage of the excellent predictive ability of the proposed EPIC framework, we estimate the changes in EPIC in the next four years along with the future revenue that will be generated by such a policy. We assume that crude oil has a heating content of 5.721 MMBtu per barrel [23] and a petroleum refinery efficiency of 90%. Moreover, the amount of petroleum and petroleum products being sent for electricity generation is assumed to be negligible (~0.57% over the last year). Crude oil demand is considered as inelastic in the short-run, with long-run values of elasticity being generally higher in absolute values but still well below 1 [25–28].

The average monthly difference (in US\$/MMBtu) and the average monthly percentage increase of EPIC in comparison to its reference values from January 2003 to September 2019 are presented in Table 2 for the different values of crude oil tax. Since crude oil is inelastic, the investigated crude oil tax has not affected the historical values of crude oil consumption. As can be seen, a US\$ 10.25 per barrel of crude oil tax increases EPIC by US\$1.022 per MMBtu or 5.60%, while a US\$ 25 per barrel of crude oil tax rises EPIC by US\$2.493 per MMBtu or 13.65%. Table 2 also summarises the amount of revenue generated from the investigated crude oil tax scenario over the same period (January 2003 - September 2019). The average annual revenue from a US\$10.25 per barrel of crude oil tax is estimated to be US\$ 71.25 billion or US\$ 17.378 billion for every US\$ 2.5 per barrel rise of crude oil tax. Supplementary Figure 4 illustrates the recalculated

EPIC for the above-mentioned values of crude oil tax along with the reference value of EPIC without tax for easy comparison, over the same period (January 2003 to September 2019).

Crude Oil	Average monthly EPIC difference	Average monthly EPIC percentage increase	Total revenue	Average annual revenue
\$/barrel	(\$/MMBtu)	(%)	(US\$ billion)	(US\$ billion)
2.5	0.249	1.37	291.083	17.378
5	0.499	2.73	582.166	34.756
7.5	0.748	4.10	873.249	52.134
10	0.997	5.46	1,164.332	69.512
10.25	1.022	5.60	1,193.440	71.250
12.5	1.246	6.83	1,455.415	86.890
15	1.496	8.19	1,746.498	104.269
17.5	1.745	9.56	2,037.581	121.647
20	1.994	10.92	2,328.664	139.025
22.5	2.244	12.29	2,619.747	156.403
25	2.493	13.65	2,910.830	173.781

We can also calculate the average increase in energy related expenses per household as a result of the proposed increase in crude oil tax using data from the relevant survey published by EIA [29] in conjunction with the EPIC findings from our previous analysis. According to the 2015 survey data, the annual energy consumption per household is 77.1 MMBtu while in 2015 the average value of EPIC is US\$18.1/MMBtu. Using this information, we estimate the average annual energy related expenses per household for 2015 to be US\$1,395.84. Taking into consideration the effects on EPIC from the increase in the crude oil taxation, a US\$2.5 per barrel increase in crude oil tax would have led to an average rise of US\$ 0.2432/MMBtu or 1.346% in EPIC for 2015. Therefore, the projected average annual energy related expenses per household would have increased by US\$18.75, up to a total of US\$1,414.59. Similarly, an increase of US\$10.25 per barrel in crude oil tax would have burdened the average energy related expenses per household by US\$77.06 or 5.5207%, up to a total of US\$1,472.90.179.

The effects on EPIC from the investigated policy during the next four years is demonstrated in Figure 4, using the predictive weights of demand of the energy products for this period. Similarly, with the results for the past period, the increase of EPIC in the future is investigated parametrically for different values of the crude oil tax.

According to Figure 4, a US\$10.25 per barrel of crude oil tax raises EPIC over the next four years by US\$0.991/MMBtu on average, while a US\$25 per barrel of crude oil tax surges EPIC by US\$2.416/MMBtu on average in the same period. Using the weights of the demand of the petroleum energy products that have been estimated from the EPIC framework along with the projections for the total annual energy demand from the EIA Annual Energy Outlook 2019 - Reference case [22], the future revenue that will be generated by the crude oil taxation policy over the next four years can be estimated. As a result, a total of US\$147.357 billion revenue is generated for every US\$5 per barrel increase in the crude oil tax over the next four years.”

Figure 4: Change in EPIC with parametric crude oil tax over the next 4 years

A new section for this particular policy case study has been added in the “Methods”, as follows:

“Policy case study 1 - Crude Oil

The main assumptions for this policy case study are:

- Crude oil has a heating content of 5.721 MMBtu per barrel [23]
- The petroleum refinery efficiency is 90%.
- The amount of petroleum and petroleum products being sent for electricity generation is assumed to be negligible (~0.57% over the last year)
- Crude oil demand is considered as inelastic in the short-run, with long-run values of elasticity being generally higher in absolute values but still well below 1 [25–28]. Since crude oil is inelastic, the investigated crude oil tax has not affected the historical values of crude oil consumption.
- The future effects (up to 2023) are assessed using the predicted values for the weights of the petroleum products applying the methodology that is described in the relevant section.
- The future annual demand as well as the future nominal weights within the petroleum sector are estimated using data from the EIA Annual Energy Outlook 2019 - Reference case [22].

The revised EPIC from the subject policy case study is estimated for the past as well as for the future as follows:

Past:

$$EPIC_{policy} = EPIC_{withoutTax} + Tax_{crude} * \frac{1}{HeatContent_{crude}} * \frac{1}{RefEff_{crude}} * \sum_{p \in P \setminus 'petprod'} W_{m, 'petprod'} \quad (5)$$

Future:

$$\Delta EPIC_{policy} = Tax_{crude} * \frac{1}{HeatContent_{crude}} * \frac{1}{RefEff_{crude}} * \sum_{p \in P \setminus 'petprod'} W_{m, 'petprod'} \quad (6)$$

where Tax_{crude} stands for the parametric crude oil tax in US\$/barrel, $HeatContent_{crude}$ stands for the heating content of crude oil, $RefEff_{crude}$ stands for the petroleum refinery efficiency, and $W_{m, 'pet prod'}$ stands for the weight of petroleum products "petprod" in month m."

[22] U.S. Energy Information Administration. Annual Energy Outlook 2019 with projections to 2050. Tech. Rep., Washington, DC, USA (2019).

[23] U.S. Energy Information Administration. Monthly Energy Review. Tech. Rep., Washington, DC, USA (December 2019).

[24] Bloomberg. Obama's US\$319 billion oil tax plan raised to US\$10.25 a barrel. <https://www.bloomberg.com/news/articles/2016-02-09/obama-s-319-billion-oil-tax-plan-raised-to-10-25-a-barrel> (Accessed: 06.27.2020) (2016).

[25] Antón, A. Taxing crude oil: A financing alternative to mitigate climate change ? Energy Policy 136, 111031 (2020).

[26] Cooper, J. C. Price elasticity of demand for crude oil: estimates for 23 countries. OPEC review 27, 1–8 (2003).

[27] Caldara, D., Cavallo, M. & Iacoviello, M. Oil price elasticities and oil price fluctuations. J. Monet. Econ.103, 1–20 (2019).

[28] Baumeister, C. & Hamilton, J. D. Structural interpretation of vector autoregressions with incomplete identification: Revisiting the role of oil supply and demand shocks. Am. Econ. Rev.109, 1873–1910 (2019).

[29] U.S. Energy Information Administration. Residential energy consumption survey for 2015. Tech. Rep., Washington, DC, USA (2015).

Action 8 – A new section is added in the "Methods" section, explaining the selection of the lookback period and the parameter estimation for the rolling horizon methodology.

"Rolling horizon methodology"

The proposed rolling horizon methodology uses data from the three previous periods in order to predict the data of interest in the current and future stages. The look back period and the parameter estimation are selected considering the forecasting errors for different schemes and lookback periods.

Energy demand is highly seasonal [23] (see Supplementary Figure 3 and Supplementary Table 4), so each month needs to be trained separately. Thus, all the schemes involve the parameter

estimation for each month individually, so as to capture the seasonality effects which are crucial for accurate forecasting.

Four different approaches for minimizing the sum of the squared error are tested over four different lookback periods (24months, 36months, 48months, 60 months), with each month being trained separately.

Approaches 1 and 2 are “Weight Based”, while approaches 3 and 4 are “Demand Based”.

The mathematical formulations are shown below for the lookback period of 36 months.

Table 1 summarises the results for the four approaches and four lookback periods.

Approach 1	Approach 2
$Err_m = \sum_{m'} (EPIC_{m'} - \widehat{EPIC}_{m'})^2$ $\widehat{EPIC}_{m'} = \sum_p C_{m',p} \widehat{w}_{m,p}$ $EPIC_m = \sum_p C_{m,p} w_{m,p}$ $\sum_p \widehat{w}_{m,p} = 1$ $\widehat{w}_{m,p} \geq 0$ $\forall m' (m' - m) = (-36) \text{ or } (-24) \text{ or } (-12)$	$Err_m = \sum_{m',p} (w_{m',p} - \widehat{w}_{m,p})^2$ $\sum_p \widehat{w}_{m,p} = 1$ $\widehat{w}_{m,p} \geq 0$ $\forall m' (m' - m) = (-36) \text{ or } (-24) \text{ or } (-12)$

Approach 3	Approach 4
$Err_m = \sum_{m',p} (D_{m',p} - a_{m,p} \cdot m' + b_{m,p})^2$ $a_{m,p} \cdot m + b_{m,p} \geq 0$ $\widehat{w}_{m,p} = \frac{a_{m,p} \cdot m + b_{m,p}}{\sum_{p'} a_{m,p'} \cdot m + b_{m,p'}}$ $\forall m' (m' - m) = (-36) \text{ or } (-24) \text{ or } (-12)$	$Err_m = \sum_p (D_{m-12,p} - (a_{m,p} \cdot D_{m-24,p} + b_{m,p} \cdot D_{m-36,p}))^2$ $a_{m,p} \cdot D_{m-12,p} + b_{m,p} \cdot D_{m-24,p} \geq 0$ $\widehat{w}_{m,p} = \frac{a_{m,p} \cdot D_{m-12,p} + b_{m,p} \cdot D_{m-24,p}}{\sum_{p'} a_{m,p'} \cdot D_{m-12,p'} + b_{m,p'} \cdot D_{m-24,p'}}$

Table 1 – Results on Prediction of Weights (Different Lookback Periods)								
	lookback_24				lookback_36			
	App1	App2	App3	App4	App1	App2	App3	App4
Min Error	0.052%	0.049%	0.064%	0.067%	0.058%	0.053%	0.051%	0.127%
Max Error	0.263%	0.221%	0.410%	0.351%	0.292%	0.228%	0.290%	0.624%
Average Error	0.106%	0.099%	0.147%	0.154%	0.121%	0.104%	0.127%	0.336%
	lookback_48				lookback_60			
	App1	App2	App3	App4	App1	App2	App3	App4
Min Error	0.064%	0.057%	0.052%	0.233%	0.064%	0.065%	0.054%	0.394%
Max Error	0.265%	0.209%	0.289%	0.776%	0.261%	0.196%	0.255%	0.839%
Average Error	0.137%	0.107%	0.120%	0.454%	0.158%	0.113%	0.115%	0.536%

As it can be seen, approaches 1 and 2 outperform approaches 3 and 4, regardless of the lookback period. Between the weight-based approaches, approach 2 is the one with the lowest errors regardless of the lookback period, so it is the one selected.

With regards to the lookback periods, the cases for 24 and 36 months produce the lowest errors in comparison to the 48 and 60 months. Since, the results for the 24 and 36 months are comparable, the longer lookback period is selected, which will capture better the increasing volatility of the energy demand in the future. Therefore, the methodology selected is Approach 2 with 36 months lookback period.

[23] U.S. Energy Information Administration. Monthly Energy Review. Tech. Rep., Washington, DC, USA (December 2019).”

REVIEWERS' COMMENTS

Reviewer #1 (Remarks to the Author):

Thanks to the authors for providing responses to my and other reviewers' concerns. The manuscript is now much improved. I have just one remaining question and request for the authors.

First, the authors say "On the contrary, EPIC captures the prices of all energy feedstocks while the weights are calculated from the actual demands of these energy feedstocks". This difference from previous methods is well taken. Could the authors include an additional Table in Supplementary Table 1 containing example weights from your analysis for easy comparison? Presumably the weights calculated by the authors vary over time, correct? If so, just a snapshot from one instance of time would be very useful for readers.

Along the same lines, some explanation about Supplementary Table 1 is required. Why are the categories all different? How should one compare the weights across different indices? Or are they not comparable at all?

The new Figure 1 is good. Thanks for updating it.

An overall comment is that as a reviewer, it is hard to go back and forth between your responses to the questions, "Actions" and the files. In the future, the authors might consider including the details about "Actions" within the responses themselves. Just a suggestion.

Reviewer #2 (Remarks to the Author):

Dear Editor,

Many thanks to give me a chance to review this article.

The authors claim that they have developed a novel framework, an index to calculate the average price of energy in the United States. The determine the total demand of the energy products directed to the end-use sectors, and the corresponding price of each product.

I have read the revised version and Action 7 and Action 8 that relates to my comments. The authors have mentioned all the details and responded the comments well.

However, i have no further objection to proceed this manuscript.

Response to Reviewers' Comments for the Revised Manuscript:

A framework to predict the price of energy for end-users with applications to the energy and monetary policies

Reviewer #1 (Remarks to the Author):

Thanks to the authors for providing responses to my and other reviewers' concerns. The manuscript is now much improved. I have just one remaining question and request for the authors.

First, the authors say "On the contrary, EPIC captures the prices of all energy feedstocks while the weights are calculated from the actual demands of these energy feedstocks". This difference from previous methods is well taken. Could the authors include an additional Table in Supplementary Table 1 containing example weights from your analysis for easy comparison? Presumably the weights calculated by the authors vary over time, correct? If so, just a snapshot from one instance of time would be very useful for readers.

Along the same lines, some explanation about Supplementary Table 1 is required. Why are the categories all different? How should one compare the weights across different indices? Or are they not comparable at all?

The new Figure 1 is good. Thanks for updating it.

An overall comment is that as a reviewer, it is hard to go back and forth between your responses to the questions, "Actions" and the files. In the future, the authors might consider including the details about "Actions" within the responses themselves. Just a suggestion.

Response: We would like to thank once again the first reviewer for the interest in our work and the constructive feedback. Please find the responses and the corresponding actions for your comments below.

Comment R1.1: *First, the authors say "On the contrary, EPIC captures the prices of all energy feedstocks while the weights are calculated from the actual demands of these energy feedstocks". This difference from previous methods is well taken. Could the authors include an additional Table in Supplementary Table 1 containing example weights from your analysis for easy comparison? Presumably the weights calculated by the authors vary over time, correct? If so, just a snapshot from one instance of time would be very useful for readers.*

Response R1.1: An additional table (Supplementary Table 3) that captures the weights of the demands of the energy products for the first 6 months of 2020 has been added. Each product has its own, unique weights and the weights of each product vary over time.

Comment R1.2: *Along the same lines, some explanation about Supplementary Table 1 is required. Why are the categories all different? How should one compare the weights across different indices? Or are they not comparable at all?*

Response R1.2: The energy indices that are presented in the Supplementary Table 1 are some of the most widely used indices in the energy sector and were used so as to highlight the difference among these indices and the proposed energy price index. These three indices as well as the energy price index differ by definition, thus their formulations are different. Moreover, the weights of these indices are different and reflect different areas of the energy sector and the financial markets. They are also completely different from the weights that are used in the energy price index. Any comparison among the indices shall be made with caution since as already mentioned they reflect different areas of the energy sector and the financial markets.

Comment R1.3: *An overall comment is that as a reviewer, it is hard to go back and forth between your responses to the questions, “Actions” and the files. In the future, the authors might consider including the details about “Actions” within the responses themselves. Just a suggestion.*

Response R1.3: We apologize for any inconvenience. Following your suggestion, we have incorporated all the details and “Actions” within our response here.

Reviewer #2 (Remarks to the Author):

Many thanks to give me a chance to review this article.

The authors claim that they have developed a novel framework, an index to calculate the average price of energy in the United States. They determine the total demand of the energy products directed to the end-use sectors, and the corresponding price of each product.

I have read the revised version and Action 7 and Action 8 that relates to my comments. The authors have mentioned all the details and responded to the comments well.

However, I have no further objection to proceed with this manuscript.

Response: We would like to thank the second reviewer for his positive feedback.